# Structural insights into modulation and selectivity of transsynaptic neurexin–LRRTM interaction

Atsushi Yamagata[1,2,3,4], Sakurako Goto-Ito[1,2,3], Yusuke Sato[1,2,3,4], Tomoko Shiroshima[1,2,3], Asami Maeda[1,2,3], Masahiko Watanabe[5], Takashi Saitoh[6], Katsumi Maenaka [7,8], Tohru Terada [9], Tomoyuki Yoshida[10,11], Takeshi Uemura[3,12,13] & Shuya Fukai [1,2,3,4]

Leucine-rich repeat transmembrane neuronal proteins (LRRTMs) function as postsynaptic organizers that induce excitatory synapses. Neurexins (Nrxns) and heparan sulfate proteoglycans have been identified as presynaptic ligands for LRRTMs. Specifically, LRRTM1 and LRRTM2 bind to the Nrxn splice variant lacking an insert at the splice site 4 (S4). Here, we report the crystal structure of the Nrxn1β–LRRTM2 complex at 3.4 Å resolution. The Nrxn1β–LRRTM2 interface involves $Ca^{2+}$-mediated interactions and overlaps with the Nrxn–neuroligin interface. Together with structure-based mutational analyses at the molecular and cellular levels, the present structural analysis unveils the mechanism of selective binding between Nrxn and LRRTM1/2 and its modulation by the S4 insertion of Nrxn.

[1] Institute for Quantitative Biosciences, The University of Tokyo, Tokyo 113-0032, Japan. [2] Synchrotron Radiation Research Organization, The University of Tokyo, Tokyo 113-0032, Japan. [3] CREST, JST, Saitama 332-0012, Japan. [4] Department of Computational Biology and Medical Sciences, Graduate School of Frontier Sciences, The University of Tokyo, Chiba 277-8561, Japan. [5] Department of Anatomy, Hokkaido University Faculty of Medicine, Sapporo 060-8638, Japan. [6] Department of Medicinal Chemistry, Faculty of Pharmaceutical Sciences, Hokkaido University of Science, Sapporo 006-8585, Japan. [7] Center for Research and Education on Drug Discovery, Faculty of Pharmaceutical Sciences, Hokkaido University, Sapporo 060-0812, Japan. [8] Laboratory of Biomolecular Science, Faculty of Pharmaceutical Sciences, Hokkaido University, Sapporo 060-0812, Japan. [9] Interfaculty Initiative in Information Studies, The University of Tokyo, Tokyo 113-0033, Japan. [10] Department of Molecular Neuroscience, Graduate School of Medicine and Pharmaceutical Sciences, University of Toyama, Toyama 930-0194, Japan. [11] PRESTO, JST, Saitama 332-0012, Japan. [12] Division of Gene Research, Research Center for Supports to Advanced Science, Shinshu University, Nagano 390-8621, Japan. [13] Department of Biological Sciences for Intractable Neurological Diseases, Institute for Biomedical Sciences, Interdisciplinary Cluster for Cutting Edge Research, Shinshu University, Nagano 390-8621, Japan. Correspondence and requests for materials should be addressed to T.U. (email: tuemura@shinshu-u.ac.jp) or to S.F. (email: fukai@iam.u-tokyo.ac.jp)

Cell adhesion molecules called synaptic organizers trigger synapse formation at the neurodevelopmental stage. Pre- and postsynaptic organizers form heterogeneous complexes across the synaptic cleft. Selective pairing between pre- and postsynaptic organizers plays important roles in assembly, establishment, and specification of neuronal synapses[1]. Neurexins (Nrxns) are a representative presynaptic organizer family and interact with several different postsynaptic organizers such as neuroligins (NLs), Cbln1–GluD2, and leucine-rich repeat transmembrane neuronal proteins (LRRTMs)[2–6]. The mammalian genome encodes three Nrxn genes[7]. Each *Nrxn* produces two isoforms, α-Nrxn and β-Nrxn. α-Nrxn has a large extracellular domain comprising six laminin/Nrxn/sex hormone-binding globulin (LNS) domains and three epidermal growth factor-like (EGF) domains, where each EGF domain is flanked by two LNS domains to form three LNS-EGF-LNS repeat units[7]. On the other hand, β-Nrxn has a small extracellular domain comprising a single LNS domain with a unique His-rich sequence at its N-terminal end[7]. The LNS domain of β-Nrxn and LNS6 of α-Nrxn are structurally equivalent and bind to the aforementioned three postsynaptic organizers. α-Nrxn has five conserved alternative splice sites, S1–S5. S4 and S5 are also contained in β-Nrxn[8]. The insertion or deletion of a 30-residue peptide at S4 (hereafter referred to as +S4 or –S4, respectively) modulates the interactions of Nrxn with NLs, Cbln1–GluD2, and LRRTMs. NLs bind to Nrxn (–S4) with a higher affinity than to Nrxn (+S4). A recent surface plasmon resonance (SPR) analysis showed that the affinity of NL1 to Nrxn (–S4) is four times higher than that to Nrxn (+S4)[9]. Similarly, previous cell-surface binding assays suggested that LRRTMs bind specifically to Nrxn (–S4)[6,10]. By contrast, Cbln1 binds specifically to Nrxn (+S4)[4]. NLs and LRRTMs bind to Nrxns in a $Ca^{2+}$-dependent manner[3,6], whereas $Ca^{2+}$ is dispensable for the binding of Cbln1 to Nrxns[11].

LRRTMs are potent postsynaptic organizers that have been shown to instruct presynaptic differentiation[12]. LRRTMs have been found only in vertebrates. Four LRRTM family genes (*Lrrtm1–Lrrtm4*) have been found in mammals[13]. Their mutations are implicated in neurodevelopmental and psychiatric disorders[14]. LRRTM1 was first isolated as a synaptogenic protein from unbiased expression screening using fibroblast–neuron co-culture assays, and the synaptogenic activities of the other three members were then assessed[12]. LRRTMs have a leucine-rich repeat (LRR) domain in the N-terminal extracellular domain, which is followed by a single transmembrane domain and a cytoplasmic region. The C-terminal of the cytoplasmic region contains a PDZ domain-binding motif. LRRTM1 and LRRTM2 can induce clustering of NMDA receptors, PSD-95, and SynGAP in the postsynaptic membrane[12], where PSD-95 functions as a scaffold to cluster these postsynaptic proteins. The C-terminal PDZ domain-binding motif of LRRTMs is essential for their binding to PSD-95[5]. All four LRRTMs can induce clustering of an excitatory synaptic marker, vesicular glutamate transporter 1 (VGLUT1), in the presynaptic terminal[12].

The isolated extracellular region of LRRTMs can instruct excitatory presynaptic differentiation[12]. The extracellular interaction between LRRTMs and Nrxn was first shown for LRRTM2 by affinity purification[5,6]. The binding of Nrxn to LRRTM2 requires a $Ca^{2+}$, similarly to the binding to NLs[6]. A cell surface-binding assay then showed that LRRTM1 can also bind to Nrxn[10]. As mentioned above, the binding to LRRTM1 and LRRTM2 was observed only for Nrxn (–S4)[6,10]. On the other hand, LRRTM4 utilizes an alternative mechanism to instruct presynaptic differentiation. Glypicans, a family of heparan sulfate proteoglycans (HSPGs), have been identified as presynaptic ligands of LRRTM4[15,16]. The LRRTM4–HSPG interaction and LRRTM4-mediated synaptic differentiation are dependent on

HS[15,16]. LRRTM3 can reportedly bind to both HSPG and Nrxn (–S4) in vitro[17]. The LRRTM3-mediated synaptogenesis is severely reduced by triple knockdown (KD) of Nrxn1–Nrxn3 but not by triple KD of glypican1, glypican2, and glypican4, suggesting that Nrxn is a primary presynaptic ligand of LRRTM3[17].

Despite the important role of the transsynaptic Nrxn–LRRTM interaction in inducing synaptic differentiation, its underlying structural mechanism remains unclear. Here, we present the crystal structures of human LRRTM2 LRR domain alone and that in complex with human Nrxn1β LNS domain at 3.15 and 3.4 Å resolutions, respectively. The complex structure between Nrxn1β and LRRTM2 unveils their $Ca^{2+}$-mediated interface, which overlaps with the Nrxn–NL interface. Together with structure-guided mutational studies by SPR analyses and fibroblast–neuron co-culture assays, the present structures elucidate the mechanism of the –S4-dependent Nrxn–LRRTM interaction and selective binding of Nrxn1β to LRRTM1 and LRRTM2.

## Results

**Structure of LRRTM2.** The structure of engineered mouse LRRTM2 LRR has been reported, although 33% of the residues were mutated to enhance the thermostability of the protein[18]. The engineered LRRTM2 showed reduced binding ability to Nrxn1β by ~50-fold[18]. Therefore, we first sought to determine the LRR structure of the nearly intact LRRTM. A transient expression system using mammalian cells produced sufficient amounts of human LRRTM2 LRR for crystallization. We consequently obtained diffraction-quality crystals of LRRTM2 (residues 34–371) with the T59L mutation (LRRTM2$^{T59L}$), which should prevent the attachment of *N*-glycan at Asn57. The crystal structure of LRRTM2$^{T59L}$ was determined at 3.15 Å resolution (Fig. 1a, Table 1) by molecular replacement using the engineered mouse LRRTM2 structure (PDB 5A5C [http://dx.doi.org/10/2210/pdb5A5C/pdb]) as the search model[18]. One asparagine residue at position 57 is located in close proximity to a crystal contact, which appears to be inhibited by the attachment of *N*-glycan at Asn57 (Supplementary Fig. 1). LRRTM2$^{T59L}$ consists of 10 LRRs (LRR1–LRR10) flanked by the N- and C-terminal caps (Fig. 1a). The interior of the convex face is stabilized by a Phe spine structure, which has also been found in other neuronal LRR proteins such as Nogo receptors, FLRTs, and Slitrks, suggesting their evolutionary relationship[19–21]. The repeat structure at the convex face is partly distorted at LRR7 and LRR8. The convex side of LRR7 forms a short helix and protrudes to LRR8. LRR8 lacks a phenylalanine residue involved in forming the Phe spine. Instead, two phenylalanine residues of the LRR7 convex side are found in the Phe spine by occupying the Phe-spine residue position in LRR8 (Fig. 1b). In the thermostabilized mouse LRRTM2, this distorted repeat is replaced by the regular repeat[18], suggesting its relevance to the stability of LRRTMs.

**Structure of Nrxn1β–LRRTM2 complex.** We next tried to determine the complex structure between Nrxn1β and LRRTM2. However, our initial attempts of their co-crystallization failed, because LRRTM2 alone was preferentially crystallized. The diffraction-quality crystals of LRRTM2 grew under the condition containing malonate or citrate, which likely behaves as a chelator of $Ca^{2+}$ and dissociates the Nrxn–LRRTM complex. In these crystals, the C-terminal caps of the adjacent LRRTM2 molecules extensively contact with each other. This contact seems to be stabilized by the intermolecular hydrophobic interaction between Trp333 and His355 (Fig. 1c), which possibly inhibits the co-crystallization of LRRTM2 and Nrxn1β. We thus replaced His355 with Ala to weaken this intermolecular interaction and confirmed by SPR analyses that LRRTM2$^{H355A}$ and LRRTM2$^{WT}$ have

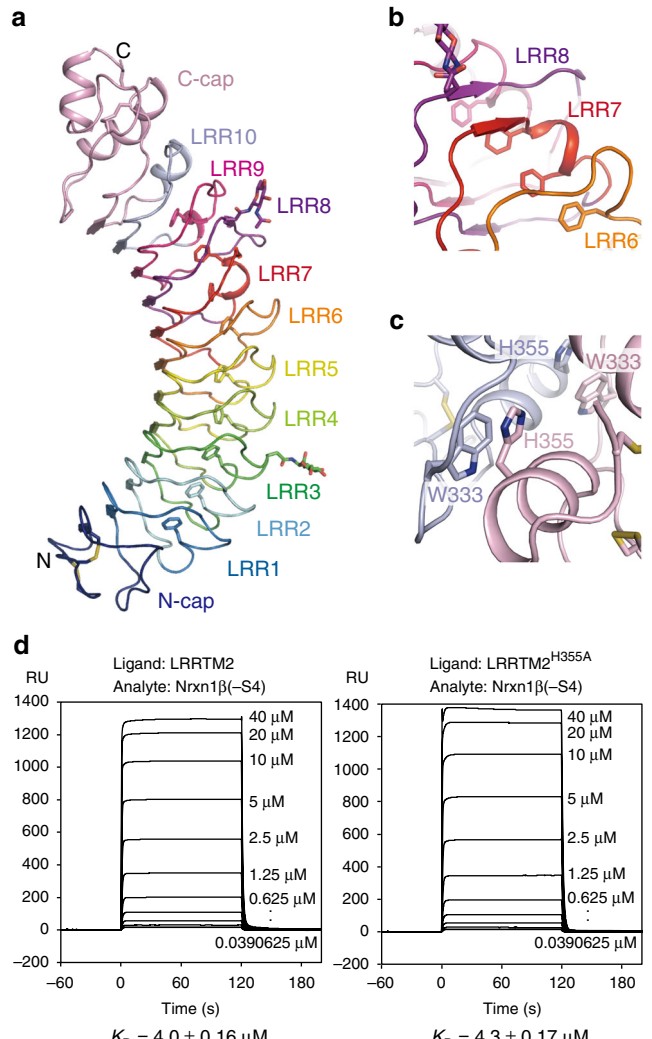

**Table 1 Data collection and refinement statistics**

| | LRRTM2[T59L] | Nrxn1β–LRRTM2[H355A] |
|---|---|---|
| Data collection | | |
| Space group | $C222_1$ | $P1$ |
| Cell dimensions | | |
| $a, b, c$ (Å) | 90.7, 240.9, 259.4 | 99.2, 99.9, 109.6 |
| $\alpha, \beta, \gamma$ (°) | 90.0, 90.0, 90.0 | 91.3, 91.1, 116.5 |
| Resolution (Å) | 50.0–3.15 (3.20–3.15)* | 50.0–3.40 (3.46–3.40)* |
| $R_{sym}$ | 0.102 (0.291) | 0.105 (0.427) |
| $I / \sigma I$ | 11.3 (1.8) | 7.2 (1.3) |
| Completeness (%) | 96.3 (88.1) | 96.3 (93.1) |
| Redundancy | 6.4 (3.2) | 3.0 (2.5) |
| Refinement | | |
| Resolution (Å) | 50.0–3.15 | 50.0–3.40 |
| No. reflections | 48,031 | 49,781 |
| $R_{work}/R_{free}$ | 0.203/0.231 | 0.209/0.241 |
| No. atoms | | |
| Protein | 10,644 | 49,781 |
| Ligand/ion | 112 | 326 |
| Water | — | 4 |
| $B$-factors (Å²) | | |
| Protein | 49.0 | 88.5 |
| Sugar/ion | 79.5 | 116.3 |
| Water | — | 59.8 |
| R.m.s. deviations | | |
| Bond lengths (Å) | 0.011 | 0.004 |
| Bond angles (°) | 1.147 | 0.890 |

*One crystal was used for each structure. *Values in parentheses are for highest-resolution shell

**Fig. 1** Structure of LRRTM2[T59L]. **a** Overall structure of LRRTM2[T59L]. The N- and C-caps and individual LRRs are colored differently. The *N*-linked sugar chains, disulfide bonds, and phenylalanine residues consisting of the Phe spine are shown as sticks. **b** Close-up view of LRR7-LRR8. Two phenylalanine residues in LRR7 (red) are involved in the Phe spine. LRR8 (purple) lacks a phenylalanine residue for the Phe spine. **c** Intermolecular interaction in LRRTM2[T59L] crystal. Two adjacent LRRTM2[T59L] molecules are colored in gray and pink. His355 in one molecule hydrophobically interacts with Trp333 in the other molecule. **d** SPR analyses of the interaction between Nrxn1β (−S4) and LRRTM2. Sensorgrams at different concentrations of Nrxn1β (−S4) are overlaid

similar affinities to Nrxin1β (−S4) ($K_D$ = 4.3 μM and 4.0 μM, respectively; Fig. 1d). We finally succeeded in the co-crystallization of Nrxn1β and LRRTM2[H355A] under the condition without malonate or citrate. The Nrxn1β–LRRTM2[H355A] complex structure was determined at 3.4 Å resolution (Table 1). Nrxn1β and LRRTM2[H355A] form a 1:1 stoichiometric complex (Fig. 2a). By analogy with protein–protein interactions mediated by other neuronal LRR proteins[21–24], Nrxn was expected to bind to the concave surface of LRRTM2[18]. On the basis of this assumption and sequence conservation, a previous computational docking analysis predicted that LRRTM2 might interact with Nrxn1β via the concave surface formed by LRR1–LRR5 and a part of the N-terminal cap[18]. However, Nrxn1β binds to the C-terminal cap of LRRTM2 with a buried surface area of 448 Å² in the present Nrxn1β–LRRTM2 structure (Fig. 2a). A calcium ion is

coordinated by the side chains of Asp141 and Asn212 and the main chains of Val158 and Ile210 in Nrxn1β (residue numbering is based on human Nrxn1β (−S4); Fig. 2b). Although the side chain of LRRTM2 Glu348 is 3.8 Å apart from the calcium ion, a residual density that could be assigned as a coordinated water molecule was observed between them. We thus modeled a water molecule in the density. The *B*-factor value of this water molecule (59.8 Å²; averaged over four molecules in the asymmetric unit; Table 1) is comparable to those of its surrounding residues, indicating that the modeling of this coordinated water molecule is reasonable. Therefore, we concluded that LRRTM2 Glu348 interacts with the calcium ion via the coordinated water molecule (Fig. 2b). The comparison between the apo-LRRTM2[T59L] and Nrxn1β-bound LRRTM2[H355A] structures (rmsd of 0.42 Å for Cα atoms) indicates that no drastic conformational change of LRRTM2 occurs upon binding to Nrxn1β, except that the side chain of LRRTM2 Glu348 flips towards the coordinated Ca²⁺ (Fig. 2c). The Ca²⁺-mediated interaction between Nrxn1β and LRRTM2 was assessed by SPR analysis of the LRRTM2 E348Q and Nrxn1β D141A mutants (Fig. 2d). The LRRTM2 E348Q and Nrxn1β D141A mutations completely and almost completely abolished the binding to Nrxn1β and LRRTM2, respectively (Fig. 2d). These findings indicate that the Ca²⁺-mediated interaction is critical for the binding between Nrxn1β and LRRTM2.

Apart from the Ca²⁺-mediated interaction, a hydrogen bond is formed between Nrxn1β Arg206 and LRRTM2 Asp352. The D352A mutation of LRRTM2 decreased the affinity by >9-fold (Fig. 2d), whereas the R206A mutation of Nrxn1β decreased it by >40-fold (Fig. 2d). The Nrxn1β–LRRTM2 interface also involves hydrophobic interactions. Phe357 of LRRTM2 hydrophobically interacts with Leu208 of Nrxn1β. Both the LRRTM2 F357A and Nrxn1β L208A mutations decreased the affinities to unmeasurable levels (Fig. 2d). Ala355 of LRRTM2[H355A],

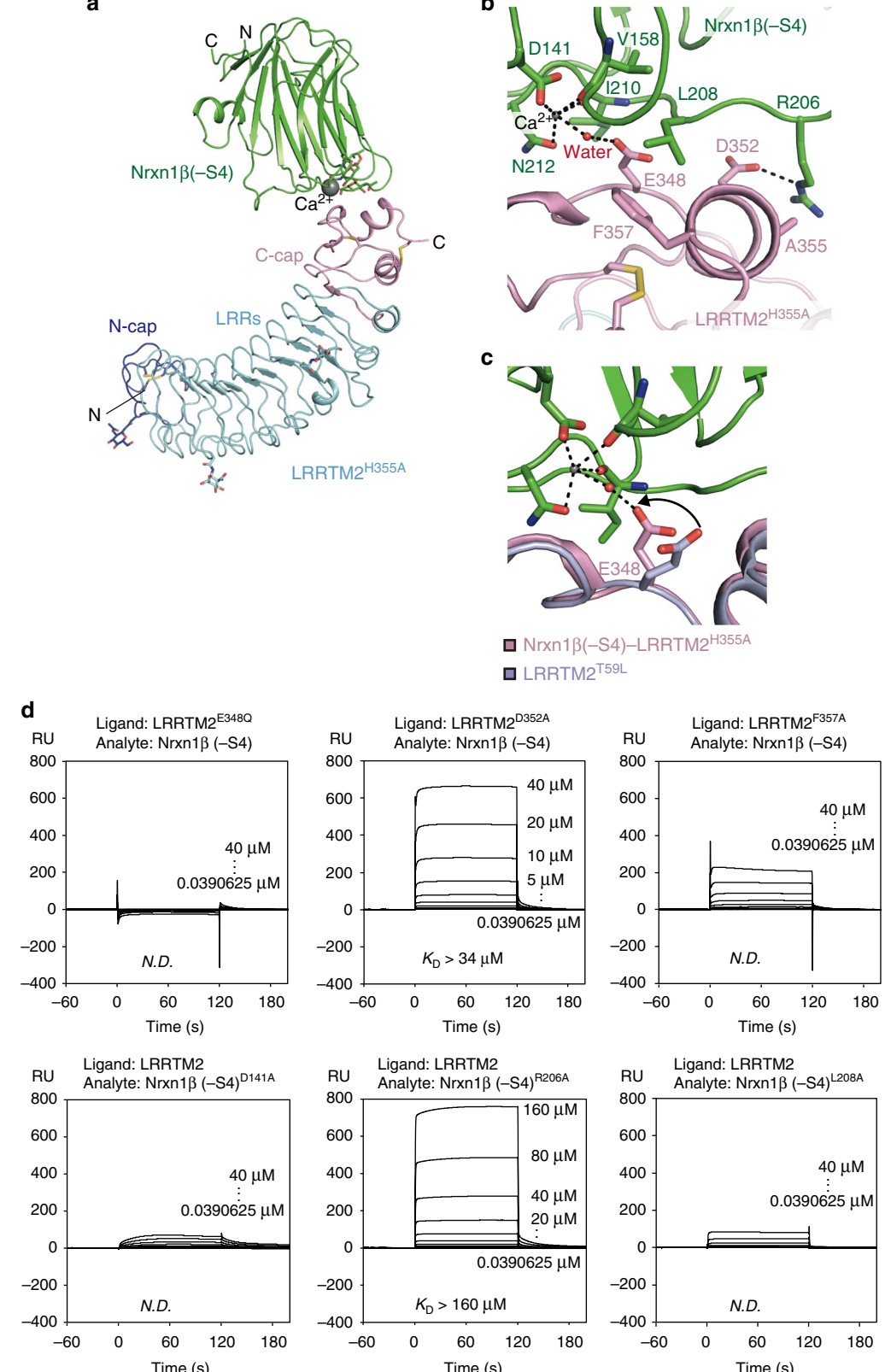

which was mutated from His355 for its co-crystallization with Nrxn1β, is located near the binding interface but is not involved in the Nrxn1β–LRRTM2 interaction in agreement with the observation that this mutation did not affect the binding (Figs 1d, 2b).

**Comparison with the interaction between Nrxn and NL**. To date, the crystal structures of the Nrxn1β–NL1 and Nrxn1β–NL4 complexes have been reported[25–27]. When the Nrxn1β–LRRTM2 complex is aligned with the Nrxn1β–NL1 and Nrxn1β–NL4 complexes so as to place the bound Nrxn1β in a similar

**Fig. 2** Structure of Nrxn1β–LRRTM2[H355A] complex. **a** Overall structure of Nrxn1β (−S4)–LRRTM2[H355A] complex. Nrxn1β (green) interacts with the C-terminal cap of LRRTM2 (C-cap; pink) but not with either LRRs (cyan) or the N-terminal cap (N-Cap; blue). Ca[2+] at the binding interface is shown as a gray sphere. **b** Close-up view of the interface between Nrxn1β (−S4) and LRRTM2[H355A]. The residues involved in the interface are shown as sticks. Hydrogen bonds and Ca[2+] coordination are shown as dotted lines. The coloring scheme is the same as that in **a**. **c** Conformational change of the side chain of LRRTM2 Glu348 upon binding to Nrxn1β. The structures of apo-LRRTM2[T59L] (light purple) and the Nrxn1β–LRRTM2[H355A] complex (pink) are superposed. **d** SPR analyses of the interaction between wild-type Nrxn1β (−S4) and mutant LRRTM2 and that between mutant Nrxn1β (−S4) and wild-type LRRTM2. Sensorgrams at different concentrations of Nrxn1β (−S4) are overlaid

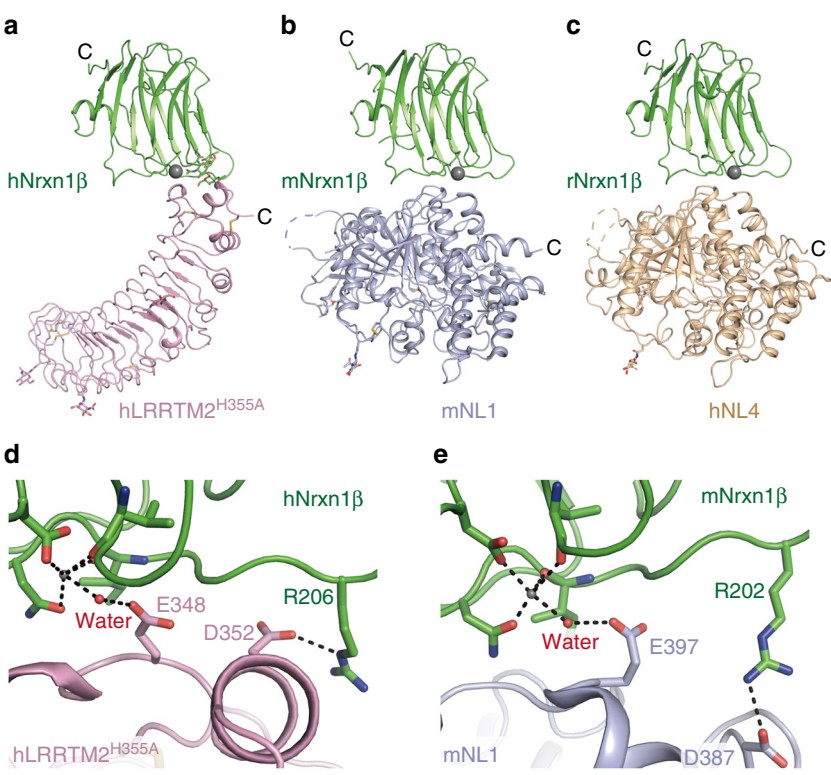

**Fig. 3** Structural comparison of Nrxn1β–LRRTM2 and Nrxn1β–NL complexes. **a** Structure of human Nrxn1β (hNrxn1β)–human LRRTM2[H355A] (hLRRTM2[H355A]) complex. Nrxn1β and LRRTM2 are colored in green and pink, respectively. The coordinated Ca[2+] is shown as a gray sphere. **b** Structure of mouse Nrxn1β (mNrxn1β)–mouse NL1 (mNL1) complex (PDB 3B3Q [http://dx.doi.org/10.2210/pdb3B3Q/pdb]). Nrxn1β and NL1 are colored in green and light purple, respectively. **c** Structure of rat Nrxn1β (rNrxn1β)–human NL4 (hNL4) complex (PDB 2WQZ [http://dx.doi.org/10.2210/pdb2WQZ/pdb]). Nrxn1β and NL4 are colored in green and beige, respectively. **d** Close-up view of the hNrxn1β–hLRRTM2[H355A] interface. The key residues in this interface are shown as sticks. The coloring scheme is the same as that in **a**. **e** Close-up view of the mNrxn1β–mNL1 interface. The key residues in this interface are shown as sticks. The coloring scheme is the same as that in **b**

orientation, the C-termini of the LRRTM2, NL1, and NL4 structures project in the same direction (Fig. 3a–c). The Nrxn1β–LRRTM2 and Nrxn1β–NL complexes appear to provide a similar spacing across the synaptic cleft. The binding interface between Nrxn1β and LRRTM2 overlaps with that between Nrxn1β and NLs, suggesting that the Nrxn1β–LRRTM and Nrxn1β–NL interactions are mutually exclusive. Indeed, previous pull-down and cell-surface binding assays have demonstrated that NL1 and LRRTM2 compete with each other for binding to Nrxn[6,10]. Our pull-down assay also showed that the presence of excess amounts of LRRTM2 inhibited the binding between Nrxn1β and NL1 (Supplementary Fig. 2a). Furthermore, in SPR analysis using Nrxn1β as an analyte and LRRTM2 as a ligand, co-injection of NL1 with Nrxn1β reduced the binding of Nrxn1β to the immobilized LRRTM2 in a dose-dependent manner (Supplementary Fig. 2b).

Nrxn binds to the acetylcholinesterase-like domain of NLs. Despite the fact that there is no structural relevance between the LRR domain of LRRTM2 and the acetylcholinesterase-like domain of NLs, they exhibit a surprising similarity in key

interactions (Fig. 3d, e). NL1 Glu397 and NL4 Glu361 form a water-mediated interaction with the bound Ca[2+] in a manner similar to LRRTM2 Glu348. NL1 Asp387 and NL4 Asp351 are functionally equivalent to LRRTM2 Asp352 for hydrogen bonding with rat/mouse Nrxn1β Arg202 (corresponding to human Nrxn1β Arg206). The Nrxn1β–NL1 and Nrxn1β–NL4 interfaces have additional hydrogen bonds (Nrxn1β Arg109–NL1 Glu297/NL4 Glu270 and Nrxn1β Ser107–NL1 Asn400/NL4 Asn364) and bury larger surface areas ($\sim$600 Å$^2$) than the LRRTM2–Nrxn1β interface (448 Å$^2$)[25–27]. Correspondingly, NL1 and NL4 have six times and twice higher affinities to Nrxn than LRRTM2, respectively ($K_D$ = 0.7 μM, 2.5 μM, and 4.0 μM for NL1, NL4, and LRRTM2, respectively), under similar experimental conditions[9].

**Selective binding of LRRTM2 to Nrxn1β (−S4).** Our SPR analysis showed that the presence of the S4 insert in Nrxn1β decreases the affinity to LRRTM2 by >20-fold ($K_D$ = 4.0 μM for Nrxn1β (−S4) and >80 μM for Nrxn1β (+S4); Fig. 4a). This

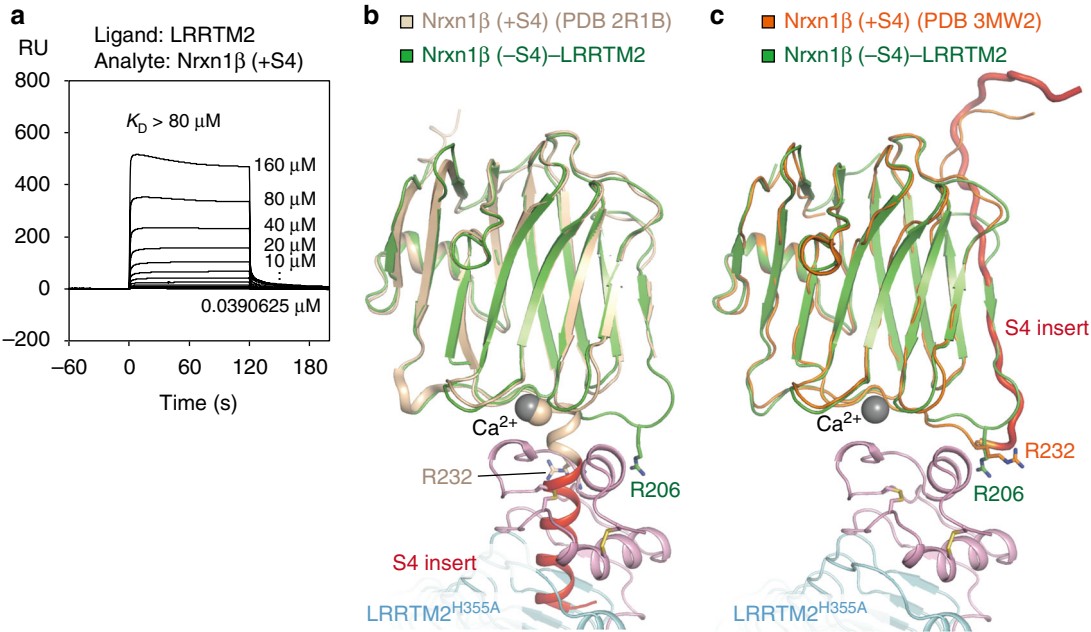

**Fig. 4** Inhibition of LRRTM2 binding by S4 insertion of Nrxn1β. **a** SPR analysis of the interaction between Nrxn1β (+S4) and LRRTM2. Sensorgrams at different concentrations of Nrxn1β (+S4) are overlaid. **b** Superposition of rat Nrxn1β (+S4) (beige; PDB 2R1B [http://dx.doi.org/10.2210/pdb2R1B/pdb]; bacterial expression) and human Nrxn1β (−S4)–LRRTM2$^{H355A}$ structures. The coloring scheme of Nrxn1β (−S4)–LRRTM2$^{H355A}$ is the same as that in Fig. 2a. The S4 insert (red) folds into an α-helix in this Nrxn1β (+S4) structure. Arg206 of human Nrxn1β (−S4) and Arg232 of rat Nrxn1β (+S4) are shown as sticks. **c** Superposition of mouse Nrxn1β (+S4) (orange; PDB 3MW2 [http://dx.doi.org/10.2210/pdb3MW2/pdb]; mammalian expression) and human Nrxn1β–LRRTM2$^{H355A}$ structures. The coloring scheme of Nrxn1β (−S4)–LRRTM2$^{H355A}$ is the same as that in **b**. The S4 insert (red) folds into a β-strand in this Nrxn1β (+S4) structure. Arg206 of human Nrxn1β (−S4) and Arg232 of mouse Nrxn1β (+S4) are shown as sticks

affinity difference between Nrxn1β (−S4) and Nrxn1β (+ S4) for LRRTM2 is larger than those for NL1 or NL4 (4-fold; $K_D$ (NL1/NL4) = 0.7 μM/2.5 μM for Nrxn1β (−S4) and 2.6 μM/9.7 μM for Nrxn1β (+S4))[9] in agreement with a previous cell-surface binding experiment[10]. The S4 insert is a 30-residue peptide positioned between Ala204 and Gly205 in human Nrxn1β. This position is immediately next to the LRRTM2-binding region of Nrxn1β. The S4 insert folds into an α-helix in rat Nrxn1β (+S4) produced in bacteria (PDB 2R1B (https://doi.org/10.2210/pdb2R1B/pdb))[28], whereas it forms a β-strand in that produced in mammalian cells (PDB 3MW2 (https://doi.org/10.2210/pdb3MW2/pdb))[29]. This conformational transition of the S4 insert can be clarified by NMR analysis, which suggested that the S4 insert is disordered in solution and likely adopts multiple conformations[30]. To assess the two different conformations of the S4 insert, Nrxn1β (+S4) produced in bacteria and that produced in mammalian cells were superposed onto the Nrxn1β–LRRTM2 complex. The α-helical conformation causes severe steric hindrance with the bound LRRTM2 (Fig. 4b), which seems to be inconsistent with the finding that Nrxn1β (+S4) retains a weak binding to LRRTM2 ($K_D$ > 80 μM). By contrast, the β-stranded conformation causes no obvious steric hindrance. However, the Arg232 side chain of rat Nrxn1β (+S4) (corresponding to Arg206 of human Nrxn1β (−S4)) is exposed to solvent so as to lose the hydrogen bond with Asp352 of LRRTM2 (Fig. 4c), explaining how the presence of the S4 insert decreases the binding affinity to LRRTM2. This idea is consistent with the finding that the affinity of LRRTM2 to Nrxn1β (+S4) ($K_D$ > 80 μM; Fig. 4a) is comparable to that to Nrxn1β (R206A) ($K_D$ > 160 μM; Fig. 2d).

**Comparison with other LRRTM family members.** Four members of the human LRRTM family are highly similar in their LRR domains with >55% sequence identity. Glu348, Asp352, and Phe357 of LRRTM2, which are critical for binding to Nrxn1β, are

completely conserved in LRRTM1 but not perfectly in LRRTM3 or LRRTM4 (Fig. 5a). Glu348 of LRRTM2 is replaced by Val in LRRTM3 (Fig. 5a). This Glu–Val replacement likely abolishes the interaction with the coordinated Ca$^{2+}$ in Nrxn1β. Phe357 of LRRTM2 is replaced by Tyr in LRRTM3 and LRRTM4. This Phe–Tyr replacement might affect the interface via the additional hydroxyl group, although both Phe and Tyr are functionally equivalent for hydrophobic or stacking interactions in general. LRRTM1 and LRRTM2 have been extensively studied in the context of the interaction with Nrxn, whereas LRRTM3 and LRRTM4 have not. We thus decided to examine the interactions of LRRTM3 and LRRTM4 with Nrxn1β by SPR analysis. As LRRTM3 was not sufficiently produced in our expression system, we constructed an LRRTM2-based LRRTM2–LRRTM3 chimera in which the Nrxn-binding region (specifically, the C-terminal region starting from the first disulfide bond in the C-terminal cap) was replaced by the corresponding region of LRRTM3 (hereafter referred to as LRRTM2+3; Fig. 5b). As expected, no binding was detected between LRRTM2+3 and Nrxn1β (Fig. 5c). Similarly, LRRTM4 also showed no binding to Nrxn1β (Fig. 5c). The F357Y mutation of LRRTM2 decreased its affinity by ~5-fold but retained its binding to Nrxn1β (Fig. 5c), suggesting that other LRRTM3/4-specific residue(s) may block the interaction of LRRTM3 and LRRTM4 with Nrxn.

**Synaptogenic activities of wild-type and mutant LRRTM2.** We then tested the effect of LRRTM2 mutations that disrupt the binding to Nrxn1β on inducing presynaptic differentiation using synaptogenic co-culture assay. When HEK293T cells expressing wild-type LRRTM2 were co-cultured with cortical neurons, the accumulation of the presynaptic active zone protein Bassoon was detected on the surface of the transfected HEK293T cells (Fig. 6a, b). On the other hand, the E348Q, D352A, and F357A mutants of LRRTM2, which abolished or reduced the binding

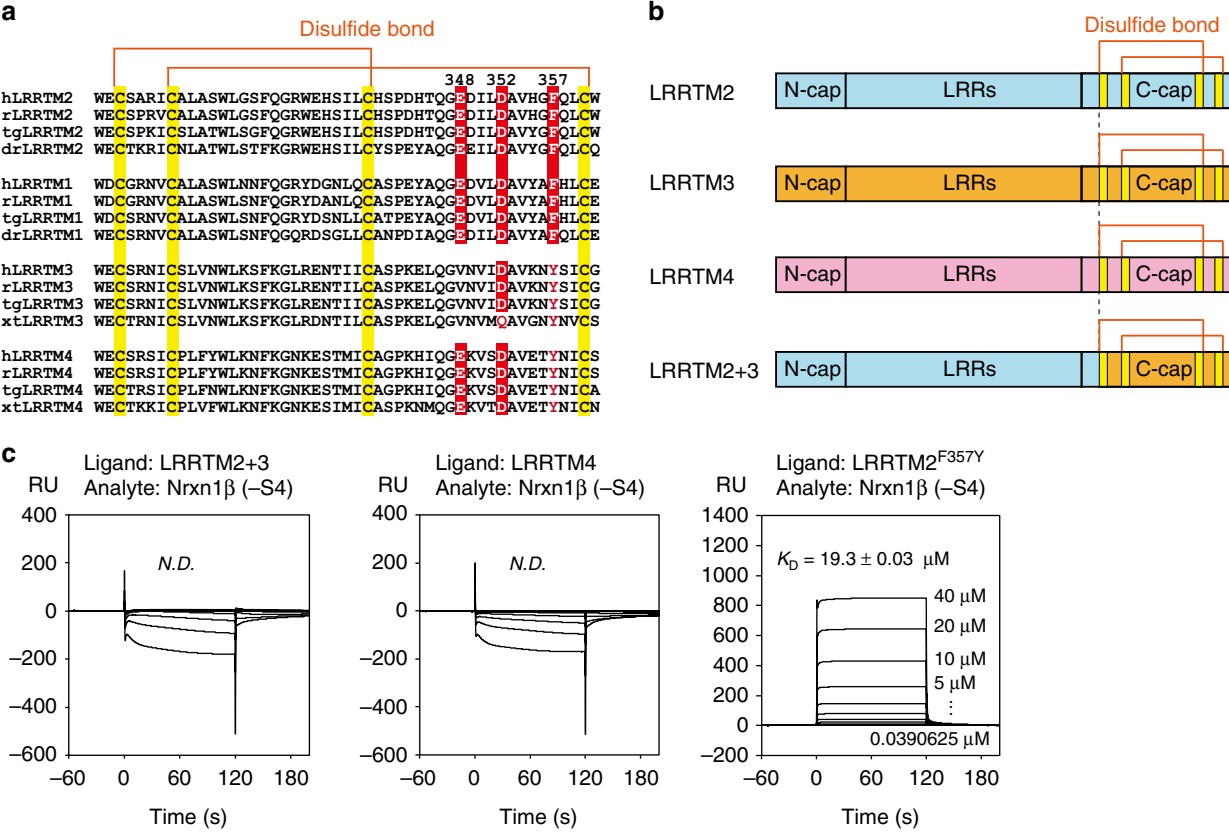

**Fig. 5** Selective binding of Nrxn1β to LRRTM1 and LRRTM2. **a** Sequence alignment of the C-terminal caps of LRRTM1–LRRTM4. Three key residues of LRRTM2 for the interaction with Nrxn1β (i.e., Glu348, Asp352, and Phe357) and the corresponding conserved residues of LRRTM1, LRRTM3, and LRRTM4 are highlighted in red with white letters. The cysteine residues forming disulfide bonds are highlighted in yellow. h, *Homo sapiens*; r, *Rattus norvegicus*; tg, *Taeniopygia guttata*; dr, *Danio rerio*. **b** Schematic diagrams of LRRTM2, LRRTM3, LRRTM4, and LRRTM2+3. LRRTM2+3 is an LRRTM2-based chimera where the C-terminal region starting at the first disulfide bond in the C-terminal cap (C-cap) of LRRTM2 is replaced by the corresponding region of LRRTM3. **c** SPR analyses of the interactions of Nrxn1β (–S4) with LRRTM2+3, LRRTM4, and LRRTM2^F357Y. Sensorgrams at different concentrations of Nrxn1β (–S4) are overlaid

affinity to Nrxn1β, did not or only slightly induced presynaptic differentiation. By contrast, LRRTM2^H355A, which was used for crystallization, and LRRTM2^W333A/H355A induced presynaptic differentiation at a similar level to the wild type. Consistently, Nrxn was accumulated on the surface of HEK293T cells expressing wild-type LRRTM2 and its mutant having presynaptic inducing activity but not on the surface of those expressing mutants lacking presynaptic inducing activity (Fig. 6a–c). There was no substantial difference in the amount of the cell surface LRRTM2 protein between the wild type and the mutants in these experiments (Fig. 6a, d).

## Discussion

In a previous study, 15 point mutations of mouse Nrxn1β were analyzed regarding its binding to LRRTM2 by cell-surface binding assays[10]. Among them, 5 mutations (corresponding to D141A, N157A, R206A, L208A, and N212A of human Nrxn1β) almost abolished the binding. In the present Nrxn1β–LRRTM2 structure, Asp141 and Asn212 of Nrxn1β coordinate a calcium ion that is critical for the Nrxn1β–LRRTM2 binding, whereas Arg206 and Leu208 directly interact with LRRTM2 (Supplementary Fig. 3a). Asn157 of Nrxn1β forms a hydrogen bond with the main chain of Gly159 and likely stabilizes the conformation of the loop encompassing Val158–Asp162. The N157A mutation may affect the $Ca^{2+}$ coordination by Val158 in this loop. In addition to these 5 critical mutations, 7 other mutations (corresponding to S111A, R113A,

D162A, N188A, Q207A, T209A, and I210A of human Nrxn1β) reduced the binding to ~40% of wild-type levels. These residues are located around the interface between Nrxn1β and LRRTM2, although none of them are directly involved in the Nrxn1β–LRRTM2 interaction. Asp259 and Thr261 of LRRTM2 (assigned as Asp260 and Thr262, respectively, in a previous report) have been identified as critically important residues for the binding to Nrxn1β and synaptogenic activity[10]. These two residues are located in the concave surface of LRR9 and are distant from the interface with Nrxn1β (Supplementary Fig. 3b). We therefore attempted to reassess the binding of LRRTM2^D259A/T261A to Nrxn1β in vitro, but failed to produce LRRTM2^D259A/T261A using our expression system. Next, to examine possible effect of the D259A/T261A mutation of LRRTM2 on the interaction with Nrxn1β, we generated models of the mutant and performed molecular dynamics (MD) simulations. We conducted two independent MD runs for each of the wild-type and mutant Nrxn1β–LRRTM2 complexes and one MD run for each of the wild-type and mutant LRRTM2 alone. In all the simulations, the structure of LRRTM2 was stably maintained with average Cα rmsd values from the crystal structure being between 1.3 and 1.7 Å (Supplementary Fig. 3c). Furthermore, intermolecular interactions found in the crystal structure were also maintained during the MD simulations for the complexes (Supplementary Table 1). Although Glu348 of LRRTM2 forms a water-mediated interaction with $Ca^{2+}$ in the crystal structure, it directly interacted with $Ca^{2+}$ in the MD simulations both for the wild-type and mutant complexes. These

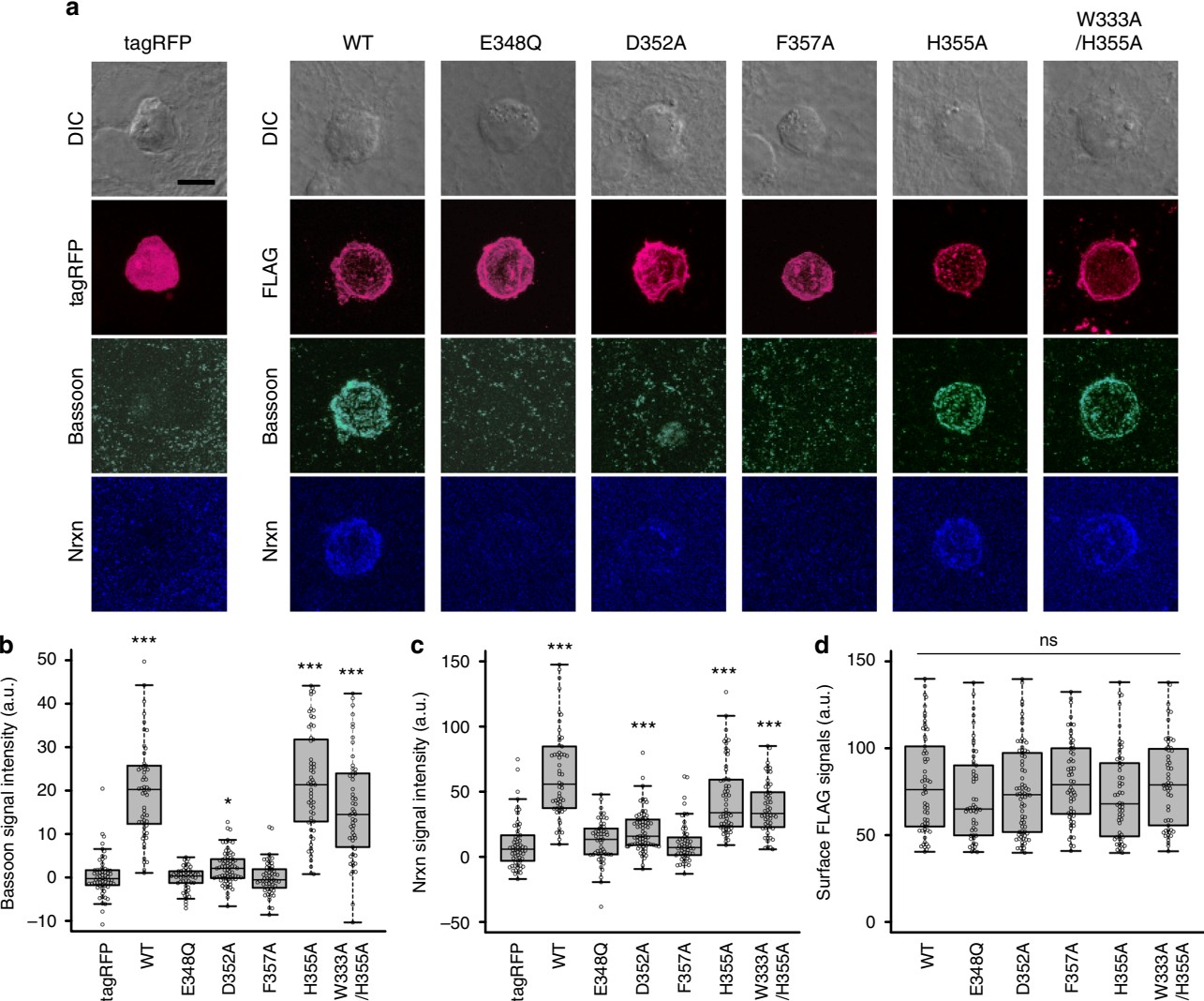

**Fig. 6** Presynapse-inducing activities of human LRRTM2 mutants. **a** Co-cultures of cortical neurons and HEK293T cells expressing TagRFP alone or FLAG-tagged human LRRTM2 mutants. Co-cultures were immunostained with antibodies against FLAG tag (magenta), Bassoon (turquoise), and Nrxn (blue). The corresponding differential interference contrast (DIC) images are also shown on the top. Scale bar represents 10 μm. **b** Staining signal intensities of Bassoon around HEK293T cells expressing FLAG-tagged LRRTM2 mutants. **c** Staining signal intensities of Nrxn around HEK293T cells expressing FLAG-tagged LRRTM2 mutants. **d** Staining signal intensities of cell surface FLAG tag on HEK293T cells expressing FLAG-tagged LRRTM2 mutants. **b–d** The horizontal line in each box indicates the median, the box shows the interquartile range (IQR), and the whiskers are 1.5 × IQR. Statistical significance was evaluated by the Kruskal–Wallis test followed by post-hoc Steel' test. ***$p < 0.001$ and *$p < 0.01$, compared with TagRFP; $n = 62$ (tagRFP), 57 (WT), 53 (E348Q), 74 (D352A), 56 (F357A), 57 (H355A), and 54 (W333A/H355A) cells

results led us to the conclusion that the D259A/T261A mutation has little effect on the interactions between LRRTM2 and Nrxn1β. Considering the secretion defect of the extracellular domain of LRRTM2$^{D259A/T261A}$ in our experiment and the result of our MD simulations, we favor the idea that the D259A/T261A mutation of LRRTM2 may affect protein folding during biosynthesis and thereby disturb its binding to Nrxn1β.

By SPR measurement, we detected no binding between LRRTM2+3 and Nrxn1β, indicating that the C-terminal cap of LRRTM3 does not bind to Nrxn1β in agreement with the structure-based sequence alignment (Fig. 5a). On the other hand, a recently reported pull-down study showed that LRRTM3 binds specifically to Nrxn1β (–S4)[17], although the binding of Nrxn1β to LRRTM3 appears to be weaker than that to LRRTM2. Technically, our SPR experiment required the addition of a low concentration of bovine serum albumin (BSA) to prevent a non-specific, irreversible binding between Nrxn1β and LRRTM2.

Previous SPR experiments of Nrxn1β by other groups have also been performed in the presence of BSA[9,29]. The discrepancy between the results of SPR and pull-down experiments of LRRTM3 and Nrxn1β might be due to their non-specific binding observed in the SPR experiment in the absence of BSA, although we cannot exclude the possibility that LRRTM3 can weakly bind to Nrxn1β in a manner different from LRRTM2. Little or no effect of the presence of BSA on our SPR analyses was confirmed by isothermal titration calorimetry (ITC) measurement, which showed a similar dissociation constant ($K_D = 4.2$ μM) in the absence of BSA (Supplementary Fig. 4). The non-specific binding of LRRTMs to Nrxns might occur when LRRTMs are immobilized on beads or sensor chips under crowded conditions.

Regarding the binding of LRRTM4 to Nrxn1β, two contradictory results have been reported: one result showed that LRRTM4 binds to Nrxn1β regardless of S4 insertion[15], whereas the other result showed no detectable interaction between Nrxn1β

and LRRTM4[16]. Our SPR measurement showed no interaction between Nrxn1β and LRRTM4, supporting the latter result. HSPG has also been identified as a potent presynaptic ligand of LRRTM4[15,16]. LRRTM3 can also bind to HSPG[17]. No detection of binding of Nrxn1β to LRRTM3 or LRRTM4 by the SPR analyses in this study raises the possibility that HSPG rather than Nrxn is a predominant ligand of LRRTM3 and LRRTM4 for inducing presynaptic differentiations.

It has been proposed that oligomerization or higher-order assembly is prerequisite for inducing synaptic differentiation mediated by several types of synaptic organizer complex: the Nrxn–NL and type IIa RPTP–SALM pairs intrinsically form a dimer[25–27,31,32], whereas the lateral assembly of the type IIa RPTP–IL1RAPL1/Slitrk pairs was deduced from their crystal packing and subsequently assessed by mutational studies on the packing interface[24,33]. Similarly, a previous study pointed out a transient oligomerization of LRRTM2[18]. Although an extensive intermolecular contact via the C-terminal cap was observed in the asymmetric unit of the apo-LRRTM2[T59L] crystal, this contact seems irrelevant to the oligomerization of LRRTM2 because the thermostabilized LRRTM2, which retains the intact C-terminal cap, does not exhibit a similar intermolecular contact in the crystal and behaves as a monomer in solution[18]. In addition, the Trp333 His355 double mutation, which should compromise the C-terminal cap-mediated intermolecular contact of LRRTM2, had no impact on synaptogenic activity (Fig. 6b). Therefore, we conclude that the intermolecular contact found in the crystal of apo-LRRTM2[T59L] is not physiologically important. Further assessment of the oligomerization or assembly of LRRTMs is needed for better understanding of their signaling mechanisms.

## Methods

**Antibodies.** The following antibodies were used: chicken anti-FLAG (Kamiya Biomedical Company; 1:500), mouse anti-Bassoon (Stressgen; RRID:AB_11181058; 1:500), rabbit anti-pan-Nrxn (RRID:AB_2571817; 0.4 μg mL$^{-1}$), which recognizes all three Nrxns[34], donkey Alexa Fluor488-conjugated anti-mouse IgG (Thermo Fisher Scientific; RRID:AB_141607; 1:500), donkey Alexa Fluor647-conjugated anti-rabbit IgG (Thermo Fisher Scientific; RRID:AB_2536183; 1:500), and donkey Cy3-conjugated anti-chicken IgY (Jackson Immuno Research; RRID:AB_2340363; 1:500).

**Cloning and plasmid construction.** The genes encoding human LRRTM2 (NM_015564.2 (https://www.ncbi.nlm.nih.gov/nuccore/194239646/)), LRRTM3 (NM_178011.4 (https://www.ncbi.nlm.nih.gov/nuccore/602617447/]), and LRRTM4 (NM_001134745.2 (https://www.ncbi.nlm.nih.gov/nuccore/1060099269/)), and Nrxn1β (–S4) (NM_001330092.1 (https://www.ncbi.nlm.nih.gov/nuccore/1052292377/)) were PCR-amplified from Human Brain Whole QUICK-CLONE$^{TM}$ cDNA (Clontech), and cloned into the pCR-Blunt II-TOPO vector (Thermo fisher Scientific). The genes encoding human LRRTM2 LRR (residues 1–371), LRRTM3 LRR (residues 1–371), and LRRTM4 LRR (residues 1–370) with the C-terminal His$_6$ tag were then cloned into the pEBMulti-Neo vector (Wako Pure Chemical Industries). The genes encoding the mouse NL1 extracellular domain (NM_001163387 (https://www.ncbi.nlm.nih.gov/nuccore/1267345270/); residues 1–666) fused with the C-terminal His$_6$ tag and the human Nrxn1β LNS domain (–S4; residues 86–266) fused with the N-terminal Igκ signal sequence and C-terminal His$_6$ tag were cloned into the pEBMulti-Neo vector. The S4 splice insert was introduced by an overlap-extension PCR method. The PCR products were then cloned into the modified pEBMulti-neo vector containing the C-terminal His$_6$ tag using a Gibson assembly cloning kit (New England Biolabs). Point mutations were introduced using a standard PCR technique. For LRRTM2+3 chimera construction, the gene fragment containing LRRTM2 (1–314) and that containing LRRTM3 (316–371) with the annealing sites were fused and cloned into the pEBMulti-neo vector using a Gibson assembly cloning kit (New England Biolabs). For pull-down assay, the gene encoding Nrxn1β (–S4; residues 85–266) was cloned into the pGEX-6P-1 vector (GE healthcare). For co-culture assay, the genes encoding wild-type and mutant LRRTM2 proteins (residues 34–516) fused N-terminally to the pre-protrypsin signal sequence and FLAG tag were cloned into the pEB6-CAG-MCS vector[35]. All primer sequences used in this study are shown in Supplementary Data 1.

**Protein preparation.** All proteins were transiently expressed in Expi293F cells (Thermo Fisher Scientific) using PEI MAX (Polyscience). For the preparation of

samples for crystallization, the C-terminally His$_6$-tagged LRRTM2 LRR was coexpressed with the C-terminally His$_6$-tagged Nrxn1β (–S4) LNS. At 7 days post-transfection, the culture media were collected and applied onto a Ni-NTA (Qiagen) column. After washing with 20 mM Tris–HCl buffer (pH 8.0) containing 300 mM NaCl and 30 mM imidazole, the proteins were eluted with 20 mM Tris–HCl buffer (pH 8.0) containing 300 mM NaCl and 250 mM imidazole. LRRTM2 LRR and Nrxn1β (–S4) LNS were separated by SEC using an ENrich SEC 650 column (Bio-Rad) with 20 mM Hepes-NaOH buffer (pH 7.5) containing 150 mM NaCl. The purified LRRTM2 LRR and Nrxn1β (–S4) LNS were concentrated using Amicon Ultra-15 30,000 MWCO and 10,000 MWCO filters (Millipore), respectively. For preparation of the samples for SPR, ITC, and pull-down experiments, LRRTM2 LRR, LRRTM2 + 3 LRR, LRRTM4 LRR, NL1 acetylcholinesterase-like, and Nrxn1β LNS domains were individually expressed in Expi293F cells and purified by Ni-affinity chromatography and SEC similarly to the samples for crystallization. The GST-fused Nrxn1β (–S4) LNS was expressed in Rosetta (DE3) *Escherichia coli* cells and purified by a Glutathione Sepharose FF column (GE healthcare) and a HiTrap Q anion-exchange column (GE healthcare).

**Crystallization.** Crystallization was carried out by the sitting-drop vapor diffusion method at 20 °C by mixing a protein solution and a crystallization solution in a 1:1 (v/v) ratio. LRRTM2$^{T59L}$ was concentrated to 2 g L$^{-1}$ and crystallized in 18% PEG3350, 0.1 M ammonium sulfate, and 0.1 M sodium malonate (pH 5.0). LRRTM2$^{H355A}$ and Nrxn1β were mixed at a 1:2 molar ratio (20 μM LRRTM2$^{H355A}$ and 40 μM Nrxn1β). CaCl$_2$ (2 mM) was supplemented for complex formation. Prior to crystallization, neuraminidase (New England Biolabs) was added at a ratio of 1:500 (v/v) to trim *N*-linked glycans. The Nrxn1β–LRRTM2$^{H355A}$ complex was crystallized in 15% PEG3350 and 0.1 M sodium/potassium phosphate (pH 6.8). The crystals were soaked in the crystallization solutions supplemented with 25% ethylene glycol and then flash-frozen in liquid N2.

**Structure determination.** Diffraction data sets were collected at 100 K at BL41XU in SPring-8 and processed with HKL2000[36] and the CCP4 program suite[37]. The LRRTM2$^{T59L}$ structure was determined by molecular replacement with the program Molrep[38] using the engineered mouse LRRTM2 structure (PDB 5A5C (https://doi.org/10.2210/pdb5A5C/pdb)) as the search model. The asymmetric unit contained four LRRTM2$^{T59L}$ molecules. The atomic model was initially built using the Phenix AutoBuild wizard[39], and then manually improved using the program Coot[40]. The structure refinement was performed using the program Phenix[39]. The Nrxn1β–LRRTM2$^{H355A}$ complex structure was determined by molecular replacement using the LRRTM2$^{T59L}$ and Nrxn1β (PDB 3B3Q (https://doi.org/10.2210/pdb3B3Q/pdb)) structures as the search models with the program Phaser[41]. The asymmetric unit contained four Nrxn1β–LRRTM2$^{H355A}$ complexes. The atomic model was manually built using the program Coot[40]. The structure refinement was performed using the program Phenix[39]. Stereochemistry was assessed by the program MolProbity[42] (94.3% favored, 0.0% outliers for LRRTM2$^{T59L}$; 95.6% favored, 0.0% outliers for Nrxn1β–LRRTM2$^{H355A}$). The final model of the Nrxn1β–LRRTM2$^{H355A}$ complex includes one calcium ion and one water molecule at the binding interface. A zinc ion was found at the crystal contact between the two adjacent LRRTM2 molecules. Data collection and refinement statistics are summarized in Table 1. The buried surface area was calculated using the program PISA[43]. All structure figures were prepared using the program PyMol (Schrödinger, LLC).

**SPR analysis.** All proteins examined by SPR analyses were purified by Ni-affinity chromatography and SEC to high homogeneity (Supplementary Fig. 5). SPR experiments were performed using Biacore T200 (GE Healthcare) at 25 °C in 20 mM Hepes-NaOH buffer (pH 7.5) containing 150 mM NaCl, 2 mM CaCl$_2$, 0.005 % Tween-20, and 0.5 g L$^{-1}$ BSA. Wild-type or mutant LRRTM2 was immobilized on a CM5 sensor chip by the amine-coupling method. The amount of the immobilized ligand for each experiment is shown in response units (RU) in Supplementary Fig. 6. The wild-type or mutant Nrxn1β (–S4) except for R206A was prepared in a two-fold serial dilution series from 40 μM. Nrxn1β ( + S4) or Nrxn1β (–S4; R206A) was prepared in a two-fold serial dilution series from 160 μM. Each sample was injected in order of increasing concentration for 120 s at a flow rate of 30 μL min$^{-1}$, followed by a 600-sec dissociation phase. The LRRTM2-immobilized sensor chip was regenerated by 20 mM Hepes-NaOH (pH 7.5) buffer containing 0.1 M EDTA and 1 M NaCl. Equilibrium dissociation constants ($K_D$) were calculated using Biacore T200 software (Supplementary Fig. 6). Data are shown as mean ± standard deviation from three independent experiments for each sample. For competition assay, Nrxn1β (–S4):NL1 mixture at a 1:1 or 1:2.5 molar ratio was prepared in a two-fold serial dilution series from 40 μM Nrxn (–S4) and 40 μM or 100 μM NL1. Each sample was injected in order of increasing concentration for 120 s at a flow rate of 30 μL min$^{-1}$, followed by a 300-sec dissociation phase.

**ITC analysis.** The purified LRRTM2 and Nrxn1β (–S4) were dialyzed against 20 mM Hepes-NaOH buffer (pH 7.5) containing 150 mM NaCl, 2 mM CaCl$_2$, and 0.005% Tween-20. This dialysis process was repeated twice. ITC experiments were performed at 25 °C using MicroCal Auto-iTC (GE Healthcare). LRRTM2 (15 μM) was placed in the cell, and Nrxn1β (200 μM) was added to the cell in a series of

injections. Binding constant was calculated by fitting the collected data with a one-site binding model using the software Origin (MicroCal).

**Pull-down assay**. Purified LRRTM2 (5 μM) or NL1 (5 μM) was mixed with GST-Nrxn1β (–S4) (5 μM) in the binding buffer containing 20 mM Hepes-NaOH (pH 7.5), 150 mM NaCl, 2 mM CaCl$_2$, and 0.005% Tween-20, and then immobilized onto Glutathione Sepharose FF (GE Healthcare) beads. For competition assay, NL1 (5 μM), GST-Nrxn1β (–S4), and LRRTM2 were mixed at a 1:1:1 or 1:1:5 molar ratio, and then immobilized onto Glutathione Sepharose FF (GE Healthcare) beads. The beads were washed with the binding buffer three times. The bound protein complexes were then eluted by SDS sample loading buffer and the eluate was subjected to SDS-PAGE with Coomassie brilliant blue staining.

**Cell cultures and co-culture assay**. Primary cortical cultures were prepared from mice at E18. The cortical cells were placed on coverslips coated with 30 μg mL$^{-1}$ poly-L-lysine and 10 μg mL$^{-1}$ mouse laminin at a density of $5 \times 10^5$ cells per well for co-culture assay. The cells were cultured in Neurobasal-A supplemented with 2% B-27 supplement (Thermo Fisher Scientific), 5% FCS, 100 U mL$^{-1}$ penicillin, 100 μg mL$^{-1}$ streptomycin, and 0.2 mM GlutaMax-I (Thermo Fisher Scientific) for 24 h and then cultured in the same medium without FCS. Cultures of HEK293T cells[44] were maintained in DMEM supplemented with 10% FCS. Expression vectors for TagRFP (pTagRFP-C; Evrogen) and FLAG-tagged wild-type and mutant hLRRTM2 proteins were transfected to HEK293T cells using X-tremeGENE HP DNA transfection reagent (Sigma-Aldrich). After 2 days of culture, the transfected cells were washed with PBS containing 2 mM EDTA and incubated with the same buffer at 37 °C for 10 min. The cells were dispersed and plated onto cortical neurons at days in vitro (DIV) 8. After 24 h of co-culture, the cells were fixed with 4% paraformaldehyde/4% sucrose and incubated with a chicken anti-FLAG antibody (1:500) for cell surface staining. After washing, cells were incubated with PBS buffer containing 0.25% Triton-X and immunostained with antibodies against Bassoon (1:500) and Nrxn (0.4 μg mL$^{-1}$), followed by incubation with donkey Alexa Fluor 488-conjugated anti-mouse IgG (1:500), donkey Cy3-conjugated anti-chicken IgY (1:500), and donkey Alexa Fluor 647-conjugated anti-rabbit IgG (1:500). Animal care and experimental protocols were reviewed by the Committee for Animal Experiments and approved by the president of Shinshu University (Authorization No. 280017 and 290072), and conducted in accordance with the Guidelines for the Care and Use of Laboratory Animals of Shinshu University.

**Image acquisition and quantification**. Images of fibroblast–neuron co-culture experiments were taken with a confocal laser-scanning microscope (TCS SP8; Leica Microsystems) under constant conditions in terms of laser power, pinhole size, gain, z-steps, and zoom setting throughout the experiments. The images were collected from at least two separate cell cultures. All quantitative measurements were performed with ImageJ 1.47 v software[45]. The intensities of immunostaining signals for Bassoon, FLAG, and Nrxn were measured as the mean florescence intensity within circles of 30 μm diameter enclosing transfected HEK293T cells.

**Statistical analysis**. Results of at least two independent experiments were subjected to statistical analyses. No statistical method was used to determine sample size. No data were excluded. There was no randomization of samples before analysis. Shapiro-Wilk and Levene's tests were used to assess the assumptions of normality and homogeneity of variance, respectively. Statistical significance was evaluated by the Kruskal–Wallis test followed by post-hoc Steel's test using R software (R Core Team, 2017).

**MD simulation**. Models of the D259A/T261A mutant of LRRTM2 were generated by removing Cγ, Oδ1, and Oδ2 of Asp259 and Cγ and Oδ of Thr261 from the structures of LRRTM2 in the crystal structures of LRRTM2 alone and in complex with Nrxn1β. The models of the wild-type and mutant LRRTM2 alone and the wild-type and mutant Nrxn1β–LRRTM2 complexes were immersed in cubic boxes of water. The N- and C-terminals of the protein chains were blocked with an acetyl group and an *N*-methyl group, respectively. The sides of the cubes were 117 Å for LRRTM2 alone and 135 Å for the complex. Chloride ions were included to neutralize the system. Amber ff14SB force field parameters[46] were used for the proteins and the TIP3P model[47] was used for water. After energy minimization and equilibration, production MD runs were performed for 200 ns. During the simulation, the temperature was kept at 300 K using the velocity-rescaling method[48], and the pressure was kept at $1.0 \times 10^5$ Pa using the Berendsen weak coupling method[49]. Bond lengths involving hydrogen atoms were constrained using the LINCS algorithm[50,51] to allow the use of a large time step (2 fs). Electrostatic interactions were calculated with the particle mesh Ewald method[52,53]. All MD simulations were performed with Gromacs 2018[54], with coordinates recorded every 10 ps.

## Data availability
Data supporting the findings of this manuscript are available from the corresponding authors upon reasonable request. The coordinates and structure factors of apo-LRRTM2$^{T59L}$ and the Nrxn1β–LRRTM2$^{H355A}$ complex have been deposited in the Protein Data Bank with the accession codes 5Z8X and 5Z8Y, respectively.

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

## Acknowledgements

We thank the beamline staffs of BL-1A of Photon Factory (Tsukuba, Japan) and BL41XU of SPring-8 (Hyogo, Japan) for technical help during data collection. This work was supported by Platform Project for Supporting Drug Discovery and Life Science Research (Basis for Supporting Innovative Drug Discovery and Life Science Research (BINDS)) from AMED (JP18am0101093 and JP18am0101107) and Hokkaido University Biosurface project. This work was supported by grants from JSPS/MEXT KAKENHI (JP16H04749 to A.Y., JP25290021 to T.U., JP25293057 to T.Y. and JP24247014 to S.F.), JST PRESTO to T.Y., and JST CREST (JPMJCR12M5) to T.U. and S.F.

## Author contributions

A.Y. and A.M. performed sample preparation and crystallization. A.Y., S.G.-I., Y.S., and S.F. collected diffraction data. A.Y. and S.F. analyzed the collected data and determined the structures. A.Y. and T.Sh. performed SPR measurements. T.Sa. and K.M. performed ITC measurements. T.Y., T.Sh., and T.U. performed cell biological experiments. M.W. prepared a rabbit anti-pan-Nrxn antibody. T.T. performed MD simulations. A.Y. and S. F. wrote the paper with editing by U.T., and A.Y., T.U., and S.F. designed the study. S.F. supervised the study.

## Additional information

**Competing interests:** The authors declare no competing interests.

