## [Peer Review File · Nature Communications]

Reviewers' comments:

Reviewer #1 (Remarks to the Author):

The manuscript by Yamagata and colleagues reports the first crystal structure of the Nrnx1 β -LRRTM2 complex at 3.2Å resolution. Whereas the structure is novel and the complex is important to explain the structural basis of the biological foundations of this interactions, overall the manuscripts suffers from several shortcomings.

Major issues:

The language needs serious revision from a proficient English speaker. Sentences throughout the manuscript are unclear (e.g. the first sentence of the discussion), and even when the meaning is obvious, the syntax is sometimes problematic.

As this is primarily a structural biology paper, the quality (purity, aggregation, etc.) of the proteins should be shown in discussed in detail as the interpretation of the data is fundamentally based on the quality of the starting material. Having a rich documentation of Coomassie gels, size exclusion chromatography curves, etc. is paramount. For example, the fact that LRRTM2+3 does not bind to NRXN may depend on its folding, on potential aggregation, etc., and it may or may not have an actual biological meaning. Also, all other mutants tested in this work must have been purified for SPR or other experiments, why not show the relative Coomassies?

Another serious problem is the apparent "irreversible" interaction between neurexin and LRRTM2. The authors mention that just by adding BSA to the running buffer the problem was solved. First, what is the nature of the interaction? Is it a non-specific binding to chip or is it a bona fide NX-LRRTM2 interaction? If this were to be the case, it would be an interesting aspect worth some investigation. Second, how would BSA cure that irreversible interactions? Perhaps a different technology (e.g. ITC) would be in order to clarify KDs, and in solution behavior. Either way this problem should be addressed in depth.

How was the KD from SPR experiments calculated, by using the Biacore software or by the saturation curve? If by saturation curve, please show the plot and relative stats (Bmax, R2, etc).

Reviewer #2 (Remarks to the Author):

I find this is a well-written manuscript which presents first structural insights into an important synaptic receptor-ligand interaction. It is clear that the data were not trivial to generate. I recommend publication in Nat Communications with some revisions.

Summary of the key results: the authors present crystal structures of LRRTM2 and its complex with the important synaptic receptor neurexin. Crystallisation of the complex required structure-based protein engineering in order to weaken the crystallisation properties of LRRTM2 on its own. The authors checked that the mutated protein still has binding properties that are similar to the wild type. Based on the structure, the authors made functional predictions: a) that the neurexin-LRRTM and Neurexin-neurologin interactions will be mutually exclusive, and b) that not all LRRTM homologues will be able to bind neurexin efficiently. Whilst the second hypothesis is validated (at least to some extent) by SPR, the first is not formally validated. An elegant co-culture assay is used to asses if disruption of the neurexin-binding site on LRRTM2 reduces synaptogenesis, which is presumably dependent on neurexin-LRRTM interaction.

Points to be addressed:

SPR:

* It would be valuable if the authors could formally show that LRRTM and neuroligin compete for neurexin-binding, and that a 3-way complex cannot form.

* Please show fitted curves and state Rmax for each SPR experiment in the supplement. Please also indicate on these figures how many units of ligand were immobilised in each experiment. The reader should not have to accept the results without seeing these important details.

Cell assay:

* In the co-culture assay, staining for neurexin should be provided to show that bassoon accumulation correlates with neurexin accumulation. This will help exclude the possibility that LRRTM2 functions through another unknown partner in these neurons using the same binding surface.

Reviewer #3 (Remarks to the Author):

Yamagata and colleagues present the crystal structure of LRRTM2 and the neurexin-LRRTM2 complex. LRRTM2 is an important synapse organizing protein implicated in excitatory synapse development and plasticity. Though the structure of an engineered version of LRRTM2 was published before (1), this study has succeeded in determining the structure of native LRRTM2. Moreover, the co-crystal of neurexin-LRRTM2 complex is new and would be of broad interest to the field. One of the clear strengths of the study is that it helps reconcile how structurally disparate domains of neuroligins (acetylcholinesterase-like) and LRRTM1/2 (leucine-rich-repeat) bind to an almost identical surface of neurexins and compete for its binding. The study goes further in demonstrating how Ca²⁺ co-ordinates the binding surface of LRRTM2 and neurexins and explains at the structural level why LRRTM2 cannot form a physiologically relevant complex with neurexins with the thirty amino-acid splice site 4 insert. Significantly, the study helps to resolve one of the discrepancies in the field; whether the ectodomain of neurexin can bind to LRRTM3 and LRRTM4. This study lays to rest that controversy in supporting one of the studies showing that the ectodomain of neurexin (-S4) can bind to LRRTM1/2 but not LRRTM3/4, in agreement with one of the studies (2).

However, despite the significance of the study, there were two glaring omissions in this study and should only be considered for publication if these issues are resolved and addressed in depth; 1) D260, T262 in the ectodomain of LRRTM2 were found to be critically important for the binding of LRRTM2 to neurexin and for its synaptogenic activity (3). The importance of these residues in neurexin binding and synaptogenic function were further supported by other studies (4, 5). The decision by the authors to ignore this body of literature will generate more controversy and not help the field to reconcile the discrepancies. Though Yamagata and colleagues found a different set of residues to be involved in the binding interface of LRRTM2 and neurexin, how are D260, T262 involved in this interaction? Does changing these residues to alanines alter the binding interface itself? The authors should at least generate a model of the LRRTM2-neurexin and discuss the critical involvement of D260, T262 in this interaction.

2) Kajander and colleagues (1) published a crystal structure of an engineered LRRTM2 and proposed that LRR1 to 5 and the N-terminal region of LRRTM2 are required for neurexin binding. This is in sharp contrast to the current study indicating that that only the c-terminal cap of LRRTM2 is required for neurexin binding. These differences should be discussed in depth and reconciled.

Other major and minor concerns are stated below.

Major concerns.

- 1) The LRRTM2-neurexin complex is proposed to span 60 Å whereas the synaptic cleft is approximately 200 Å. How does the interaction potentially influence synaptic membrane structure? This would be especially significant for understanding synaptogenesis and potentially, plasticity.
- 2) The SPR data on binding affinities between neurexins and LRRTMs does not entirely agree with

cell-surface binding studies and the synaptogenic strength. I suggest the authors to use an independent method to determine the binding affinities, such as isothermal titration calorimetry.

- 3) In the Introduction the authors state "All four LRRTMs can induce clustering of the excitatory synaptic marker, vesicular glutamate transporter 1 (VGLUT1), in the presynaptic membrane terminal, although LRRTM3 and LRRTM4 showed less potent synaptogenic activity than LRRTM1 and LRRTM2". The synaptogenic activity by the fibroblast-neuron co-culture is subjective and depends on experimental conditions. In fact, LRRTM4's synaptogenic activity was as strong as that of neuroligin1. Either correct this statement or delete this statement because it may incorrectly generate the presumption that that LRRTM3 and LRRTM4 are weakly synaptogenic.
- 4) Please provide evidence for LRRTM2 Asn57 n-glycation. Would it be possible to model n-glycation onto the LRRTM2 T59L structure and determine its potential effects on the structure?
- 5) If the Trp333 and His355 of adjacent LRRTM2 molecules contact each other, please discuss the physiological relevance of this interaction. Could this be involved in LRRTM2 dimerization or would this directly block the interaction with neurexin?
- 6) Many of the residues in neurexin required for LRRTM2 binding were reported to be required for binding in a previous study(3). Please discuss the similarities in this study with what has been reported before.
- 7) In the discussion, the authors state "...suggesting that the expression level of LRRTMs may be comparable to or higher than those of NLs to maintain the balance between the LRRTM- and NL-mediated signals." This statement is mere speculation and does not hold up to scrutiny. Do the authors have evidence to back up this supposition? In fact, neuroligins are ubiquitously expressed and may be more abundant than LRRTMs. This discussion should be modified to reflect published evidence.

Minor points and corrections

- 1) Abstract "Leucine-rich repeat transmembrane neuronal proteins (LRRTMs) have emerged as postsynaptic organizers to induce excitatory synapses." Change to "Leucine-rich repeat transmembrane neuronal proteins (LRRTMs) have emerged as postsynaptic organizers that induce excitatory synapses."
- 2) Introduction "Synaptic adhesion molecules called synaptic organizers trigger synapse formation in the neurodevelopmental stage." Change to "Synaptic adhesion molecules called synaptic organizers trigger synapse formation at the neurodevelopmental stage".
- 3) Introduction "Mammalian genome encodes three..." Change to "The mammalian genome encodes three..."
- 4) Introduction "...a unique His-rich sequence in its N-terminal end...". Change to "...a unique His-rich sequence at its N-terminal end..."
- 5) Introduction "LNS of β -Nrxn and LNS6 of..." Change to "The LNS domain of β -Nrxn and LNS6 of..."
- 6) Introduction "recent surface plasmon resonance (SPR) analysis showed that the affinity to Nrxn (-S4) is four times higher than that to Nrxn (+S4)." Change to "recent surface plasmon resonance (SPR) analysis showed that the affinity of neuroligin1 to Nrxn (-S4) is four times higher than that to Nrxn (+S4)."
- 7) Correct notations for the genes encoding LRRTMs should be presented.
- 8) Introduction "The isolated extracellular region of LRRTMs can instruct the excitatory presynaptic differentiation." Change to "The isolated extracellular region of LRRTMs can instruct excitatory presynaptic differentiation."
- 9) Results "A transient expression system using mammalian cells produced a sufficient amount of human LRRTM2 LRR for crystallization." Change to "A transient expression system using mammalian cells produced sufficient amount of human LRRTM2 LRR for crystallization."
- 10) Results "In the thermostabilized mouse LRRTM2, this distorted repeat is replaced by the regular repeat, suggesting its relevance with the stability of LRRTMs." Change to "In the thermostabilized mouse LRRTM2, this distorted repeat is replaced by the regular repeat, suggesting its relevance to the stability of LRRTMs."

References

1. Paatero A, et al. (2016) Crystal Structure of an Engineered LRRTM2 Synaptic Adhesion Molecule and a Model for Neurexin Binding. *Biochemistry* 55(6):914-926.
2. Siddiqui TJ, et al. (2013) An LRRTM4-HSPG complex mediates excitatory synapse development on dentate gyrus granule cells. *Neuron* 79(4):680-695.
3. Siddiqui TJ, Pancaroglu R, Kang Y, Rooyakkers A, & Craig AM (2010) LRRTMs and neuroligins bind neurexins with a differential code to cooperate in glutamate synapse development. *J Neurosci* 30(22):7495-7506.
4. Soler-Llavina GJ, et al. (2013) Leucine-rich repeat transmembrane proteins are essential for maintenance of long-term potentiation. *Neuron* 79(3):439-446.
5. Um JW, et al. (2016) LRRTM3 Regulates Excitatory Synapse Development through Alternative Splicing and Neurexin Binding. *Cell Rep* 14(4):808-822.

Re: manuscript NCOMMS-18-05696

Comments from Reviewer #1:

Major issues:

The language needs serious revision from a proficient English speaker. Sentences throughout the manuscript are unclear (e.g. the first sentence of the discussion), and even when the meaning is obvious, the syntax is sometimes problematic.

We have carefully read and checked the manuscript for English usage. In addition, the manuscript was further checked by a commercial English editing service for scientific papers.

In the first paragraph of Discussion in the initial manuscript, we aimed to discuss the relationship between the lengths of the synaptic cleft and extracellular regions of synaptic organizer complexes. However, as either Nr_{xn}1 β -LRRTM2 (this study) or Nr_{xn}1 β -NLs (previous studies) structure does not contain the linker region connecting the structurally ordered extracellular domain (*i.e.*, LNS, acetylcholinesterase-like, or LRR domain) and transmembrane helix, the discussion was unclear and speculative. Therefore, we removed the first paragraph of Discussion in the revised manuscript.

... the quality (purity, aggregation, etc.) of the proteins should be shown in discussed in detail as the interpretation of the data is fundamentally based on the quality of the starting material. Having a rich documentation of Coomassie gels, size exclusion chromatography curves, etc. is paramount. ... Also, all other mutants tested in this work must have been purified for SPR or other experiments, why not show the relative Coomassies?

For all protein samples examined in this study, chromatograms of size-exclusion chromatography and SDS-PAGE gels with CBB staining were shown in Supplementary Fig. 5.

Another serious problem is the apparent “irreversible” interaction between neurexin and LRRTM2. The authors mention that just by adding BSA to the running buffer the problem was solved. First, what is the nature of the interaction? Is it a non-specific binding to chip or is it a bona fide NX-LRRTM2 interaction? If this were to be the case, it would be an interesting aspect worth some investigation. Second, how would BSA cure that irreversible interactions? Perhaps a different technology (e.g. ITC) would be in order to clarify KDs, and in solution behavior. Either way this problem should be addressed in depth.

The binding between Nrxn1 β and LRRTM2 was analyzed by isothermal titration calorimetry (ITC) experiment. K_D determined by ITC in the absence of BSA is 4.2 μ M (Supplementary Fig. 4), which is equivalent to that determined by SPR spectroscopy in the presence of BSA, indicating little or no effect of the presence of BSA on our SPR analyses. We suppose that non-specific binding of LRRTMs to Nrxns might occur when LRRTMs are immobilized on beads or sensor chips under crowded conditions. This was mentioned in the second paragraph of Discussion (pg. 13).

How was the K_D from SPR experiments calculated, by using the Biacore software or by the saturation curve? If by saturation curve, please show the plot and relative stats (B_{max} , R^2 , etc).

We used Biacore T200 software (GE Healthcare) for calculation of K_D (equilibrium dissociation constant). According to this and Review #2's comments, the plot (concentration vs response units), fitting curve, and parameters (the amount of immobilized ligands, experimental R_{max} , and χ^2) were shown in Supplementary Fig. 6.

Comments from Reviewer #2:

Points to be addressed:

SPR:

** It would be valuable if the authors could formally show that LRRTM and neuroligin compete for neurexin-binding, and that a 3-way complex cannot form.*

Previous pull-down and cell-surface binding assays have demonstrated that NL1 and LRRTM2 compete with each other for binding to Nrxn (Ko *et al.*, *Neuron*, 2009; Siddiqui *et al.*, *J. Neurosci.*, 2010). Therefore, we first confirmed that the presence of excess amounts of LRRTM2 inhibited the binding between Nrxn1 β and NL1 by pull-down assay (Supplementary Fig. 2a). We also showed the competition between LRRTM2 and NL1 by demonstrating that co-injection with NL1 decreases the binding of Nrxn1 β to the immobilized LRRTM2 in SPR analysis (Supplementary Fig. 2b). These results were described in the subsection "Comparison with the interaction between Nrxn and NL" (pg. 8-9).

** Please show fitted curves and state R_{max} for each SPR experiment in the supplement. Please also indicate on these figures how many units of ligand were immobilised in each experiment.*

According to this and Review #1's comments, the plot (concentration vs response units),

fitting curve, and parameters (the amount of immobilized ligands, experimental R_{\max} , and Chi^2) were shown in Supplementary Fig. 6.

Cell assay:

** In the co-culture assay, staining for neurexin should be provided to show that bassoon accumulation correlates with neurexin accumulation.*

Neurexin was stained in the co-culture assay (Fig. 6) to show that bassoon accumulation correlates with neurexin accumulation.

Comments from Reviewer #3):

... However, despite the significance of the study, there were two glaring omissions in this study and should only be considered for publication if these issues are resolved and addressed in depth;

1) D260, T262 in the ectodomain of LRRTM2 were found to be critically important for the binding of LRRTM2 to neurexin and for its synaptogenic activity (3). The importance of these residues in neurexin binding and synaptogenic function were further supported by other studies (4, 5). The decision by the authors to ignore this body of literature will generate more controversy and not help the field to reconcile the discrepancies. Though Yamagata and colleagues found a different set of residues to be involved in the binding interface of LRRTM2 and neurexin, how are D260, T262 involved in this interaction? Does changing these residues to alanines alter the binding interface itself? The authors should at least generate a model of the LRRTM2-neurexin and discuss the critical involvement of D260, T262 in this interaction.

Asp260 and Thr262 of LRRTM2 are located in the concave surface of LRR9 and are distant from the interface with Nrnx1 β (Supplementary Fig. 3b). Therefore, we attempted to reassess the binding of LRRTM2^{D260A/T262A} to Nrnx1 β *in vitro*, but failed to produce LRRTM2^{D260A/T262A} using our expression system. The D260A and T262A mutations of LRRTM2 possibly affect protein folding and/or stability, and thereby disturb the binding to Nrnx1 β . This discussion was described in the first paragraph of Discussion (pg. 12).

2) Kajander and colleagues (1) published a crystal structure of an engineered LRRTM2 and proposed that LRR1 to 5 and the N-terminal region of LRRTM2 are required for neurexin binding. This is in sharp contrast to the current study indicating that that only the c-terminal cap of LRRTM2 is required for neurexin binding. These differences should be discussed in depth and reconciled.

Nrxn was assumed to bind the concave surface of LRRTM2 by analogy with protein-protein interactions mediated by other neuronal LRR proteins. On the basis of this assumption and sequence conservation, a previous computational docking analysis by *Kajander and colleagues* (incorrectly) predicted that LRRTM2 might interact with Nrxn1 β via the concave surface formed by LRR1–LRR5 and a part of the N-terminal cap. This was mentioned in the subsection “Structure of Nrxn1 β –LRRTM2 complex” (pg. 7).

To our knowledge, there is no experimental data that directly support the assumption that the concave surface of LRRTM2 mediates the binding to Nrxn1 β . It seems that this wrong assumption caused the inconsistency between the present crystal structure and previous computational model.

Major concerns.

1) *The LRRTM2-neurexin complex is proposed to span 60 Å whereas the synaptic cleft is approximately 200 Å. How does the interaction potentially influence synaptic membrane structure? This would be especially significant for understanding synaptogenesis and potentially, plasticity.*

In the first paragraph of Discussion in the initial manuscript, we aimed to discuss the relationship between the lengths of the synaptic cleft and extracellular regions of synaptic organizer complexes. However, as either Nrxn1 β –LRRTM2 (this study) or Nrxn1 β –NLs (previous studies) structure does not contain the linker region connecting the structurally ordered extracellular domain (*i.e.*, LNS, acetylcholinesterase-like, or LRR domain) and transmembrane helix, the discussion was unclear and speculative. Therefore, we removed the first paragraph of Discussion in the revised manuscript. Presumably, the structural unit composed of Nrxn1 β LNS–LRRTM2 LRR or Nrxn1 β LNS–NL acetylcholinesterase-like domain is located in the middle of the synaptic cleft, and the flexible linkers of Nrxn1 β and LRRTM2/NL connect the unit to the transmembrane helices on the pre- and postsynaptic membranes, respectively. However, a functional role of such spatial arrangement of the Nrxn1 β –LRRTM2 or Nrxn1 β –NL complex in the synaptic cleft is unclear.

2) *The SPR data on binding affinities between neurexins and LRRTMs does not entirely agree with cell-surface binding studies and the synaptogenic strength. I suggest the authors to use an independent method to determine the binding affinities, such as isothermal titration calorimetry.*

We determined the binding affinity between Nrxn1 β and LRRTM2 by isothermal titration calorimetry (Supplementary Fig. 4), which was comparable to that determined by SPR analysis.

3) *In the Introduction the authors state “All four LRRTMs can induce clustering of the excitatory synaptic marker, vesicular glutamate transporter 1 (VGLUT1), in the presynaptic membrane terminal,*

although LRRTM3 and LRRTM4 showed less potent synaptogenic activity than LRRTM1 and LRRTM2". The synaptogenic activity by the fibroblast-neuron co-culture is subjective and depends on experimental conditions. In fact, LRRTM4's synaptogenic activity was as strong as that of neuroligin1. Either correct this statement or delete this statement because it may incorrectly generate the presumption that that LRRTM3 and LRRTM4 are weakly synaptogenic.

The statement was deleted accordingly.

4) Please provide evidence for LRRTM2 Asn57 n-glycytion. Would it be possible to model n-glycation onto the LRRTM2 T59L structure and determine its potential effects on the structure?

Electron density corresponding to *N*-acetylglucosamin attached to Asn57 in the Nrxn1 β -bound LRRTM2^{H355A} was shown in Supplementary Fig. 1b. One asparagine residue at position 57 is located in close proximity to a crystal contact in the apo-LRRTM2^{H355A} crystal, which appears to be inhibited by the attachment of *N*-glycan at Asn57 (Supplementary Fig. 1a). This was mentioned in the subsection "Structure of LRRTM2" (pg. 6).

5) If the Trp333 and His355 of adjacent LRRTM2 molecules contact each other, please discuss the physiological relevance of this interaction. Could this be involved in LRRTM2 dimerization or would this directly block the interaction with neurexin?

The W333A H355A double mutation of LRRTM2 showed no impact on the synaptogenic activity (Fig. 6). The molecular contact of the LRRTM2 LRR domain is likely to be an artifact of crystallization. Therefore, we will not discuss the physiological relevance of the intermolecular contact of LRRTM2 LRR observed in the apo-LRRTM2^{T59L} crystal.

6) Many of the residues in neurexin required for LRRTM2 binding were reported to be required for binding in a previous study(3). Please discuss the similarities in this study with what has been reported before.

We discussed how previously examined point mutations of neurexin affected the binding to LRRTM2 in the first paragraph of Discussion (pg. 12) as follows:

"In a previous study, 15 point mutations of mouse Nrxn1 β were analyzed regarding its binding to LRRTM2 by cell-surface binding assays. Among them, 5 mutations (corresponding to D141A, N157A, R206A, L208A, and N212A of human Nrxn1 β) almost

abolished the binding. In the present Nrnx1 β –LRRTM2 structure, Asp141 and Asn212 of Nrnx1 β coordinate a calcium ion that is critical for the Nrnx1 β –LRRTM2 binding, whereas Arg206 and Leu208 directly interact with LRRTM2 (Supplementary Fig. 3a). Asn157 of Nrnx1 β forms a hydrogen bond with the main chain of Gly159 and likely stabilizes the conformation of the loop encompassing Val158–Asp162. The N157A mutation may affect the Ca²⁺ coordination by Val158 in this loop. In addition to these 5 critical mutations, 7 other mutations (corresponding to S111A, R113A, D162A, N188A, Q207A, T209A, and I210A of human Nrnx1 β) reduced the binding to ~40% of wild-type levels. These residues are located around the interface between Nrnx1 β and LRRTM2, although none of them are directly involved in the Nrnx1 β –LRRTM2 interaction. Asp260 and Thr262 of LRRTM2 have been identified as critically important residues for the binding to Nrnx1 β and synaptogenic activity. These two residues are located in the concave surface of LRR9 and are distant from the interface with Nrnx1 β (Supplementary Fig. 3b). We therefore attempted to reassess the binding of LRRTM2^{D260A/T262A} to Nrnx1 β *in vitro*, but failed to produce LRRTM2^{D260A/T262A} using our expression system. The D260A and T262A mutations of LRRTM2 possibly affect protein folding and/or stability, and thereby disturb its binding to Nrnx1 β .”

7) *In the discussion, the authors state “...suggesting that the expression level of LRRTMs may be comparable to or higher than those of NLS to maintain the balance between the LRRTM- and NL-mediated signals.” This statement is mere speculation and does not hold up to scrutiny. Do the authors have evidence to back up this supposition? In fact, neuroligins are ubiquitously expressed and may be more abundant than LRRTMs. This discussion should be modified to reflect published evidence.*

We deleted the statement because of no evidence to support it.

Minor points and corrections

1) *Abstract “Leucine-rich repeat transmembrane neuronal proteins (LRRTMs) have emerged as postsynaptic organizers to induce excitatory synapses.” Change to “Leucine-rich repeat transmembrane neuronal proteins (LRRTMs) have emerged as postsynaptic organizers that induce excitatory synapses.”*

2) *Introduction “Synaptic adhesion molecules called synaptic organizers trigger synapse formation in the neurodevelopmental stage.” Change to “Synaptic adhesion molecules called synaptic organizers trigger synapse formation at the neurodevelopmental stage”.*

3) *Introduction “Mammalian genome encodes three...” Change to “The mammalian genome encodes three...”*

- 4) Introduction “.....a unique His-rich sequence in its N-terminal end....”. Change to “.....a unique His-rich sequence at its N-terminal end...”
- 5) Introduction “LNS of β -Nrnx and LNS6 of...” Change to “The LNS domain of β -Nrnx and LNS6 of...”
- 6) Introduction “recent surface plasmon resonance (SPR) analysis showed that the affinity to Nrnx (–S4) is four times higher than that to Nrnx (+S4).” Change to “recent surface plasmon resonance (SPR) analysis showed that the affinity of neuroligin1 to Nrnx (–S4) is four times higher than that to Nrnx (+S4).”
- 7) Correct notations for the genes encoding LRRTMs should be presented.
- 8) Introduction “The isolated extracellular region of LRRTMs can instruct the excitatory presynaptic differentiation.” Change to “The isolated extracellular region of LRRTMs can instruct excitatory presynaptic differentiation.”
- 9) Results “A transient expression system using mammalian cells produced a sufficient amount of human LRRTM2 LRR for crystallization.” Change to “A transient expression system using mammalian cells produced sufficient amount of human LRRTM2 LRR for crystallization.”
- 10) Results “In the thermostabilized mouse LRRTM2, this distorted repeat is replaced by the regular repeat, suggesting its relevance with the stability of LRRTMs.” Change to “In the thermostabilized mouse LRRTM2, this distorted repeat is replaced by the regular repeat, suggesting its relevance to the stability of LRRTMs.”

The manuscript was corrected according to all these comments. We appreciate his/her suggestion of these minor points and corrections.

REVIEWERS' COMMENTS:

Reviewer #1 (Remarks to the Author):

The authors have been responsive to my critiques and have addressed them satisfactorily. I have no further comments at this time.

Reviewer #2 (Remarks to the Author):

I am happy with the corrections that have been made with respect to the comments I raised.

Reviewer #3 (Remarks to the Author):

Yamagata and colleagues have done well to address all but one very important concern of mine. They should address the significance of Asp260 and Thr262 of LRRTM2 more meaningfully and not be dismissive of multiple previous reports. They state "that D260A and T262A mutations of LRRTM2 possibly affect protein folding and/or stability, and thereby disturb the binding to Nrxn1 β ". However, others haven't observed surface trafficking deficits of this mutant. This mutant is expressed at neuronal surfaces just as well as the wild-type protein. I suspect that LRRTM2 D260A/T262A mutant of LRRTM2 may alter structure to seclude the direct binding sites. This can be easily tested in a molecular dynamics model. I think this is a very important consideration that must be fully addressed and not be dismissed in an off-hand manner.

Re: manuscript NCOMMS-18-05696A

Comments from Reviewer #3:

Yamagata and colleagues have done well to address all but one very important concern of mine. They should address the significance of Asp260 and Thr262 of LRRTM2 more meaningfully and not be dismissive of multiple previous reports. They state “that D260A and T262A mutations of LRRTM2 possibly affect protein folding and/or stability, and thereby disturb the binding to Nrnx1β”. However, others haven’t observed surface trafficking deficits of this mutant. This mutant is expressed at neuronal surfaces just as well as the wild-type protein. I suspect that LRRTM2 D260A/T262A mutant of LRRTM2 may alter structure to seclude the direct binding sites. This can be easily tested in a molecular dynamics model. I think this is a very important consideration that must be fully addressed and not be dismissed in an off-hand manner.

First, we note that the residues assigned as Asp260 and Thr262 in the previous report correspond to Asp259 and Thr261, respectively, in the NCBI Reference Sequence NP_056379.1. Throughout the manuscript, we used the residue numbering based on this reference sequence.

To examine possible effect of the D259A/T261A mutation of LRRTM2 on the interaction with Nrnx1β, we generated models of the mutant and performed molecular dynamics (MD) simulations, according to the comment from Reviewer #3. We conducted two independent MD runs for each of the wild-type and mutant LRRTM2–Nrnx1β complexes and one MD run for each of the wild-type and mutant LRRTM2 alone. In all the simulations, the structure of LRRTM2 was stably maintained with average Cα rmsd values from the crystal structure being between 1.3 and 1.7 Å, as shown in Supplementary Fig. 3c. Furthermore, intermolecular interactions found in the crystal structure were also maintained during the MD simulations for the complexes, as shown in Supplementary Table 1. Although Glu348 of LRRTM2 forms a water-mediated interaction with Ca²⁺ in the crystal structure, it directly interacted with Ca²⁺ in the MD simulations both for the wild-type and mutant complexes. These results led us to the conclusion that the D259A/T261A mutation has little effect on the interactions between LRRTM2 and Nrnx1β. Considering the secretion defect of the extracellular domain of LRRTM2^{D259A/T261A} in our experiment and the result of our MD simulations, we favor the idea that the D259A/T261A mutation of LRRTM2 may affect protein folding during biosynthesis and thereby disturb its binding to Nrnx1β.

This description was added in the first paragraph of Discussion (pg. 12–13).